# The Unresolved Problem of DNA Bridging

**DOI:** 10.3390/genes9120623

**Published:** 2018-12-12

**Authors:** María Fernández-Casañas, Kok-Lung Chan

**Affiliations:** Chromosome Dynamics and Stability Group, Genome Damage and Stability Centre, University of Sussex, Brighton BN1 9RQ, UK; M.Fernndez-Casanas@sussex.ac.uk

**Keywords:** Ultra-fine DNA bridges, chromosome segregation, sister chromatid disjunction, PICH/ERCC6L, Bloom’s syndrome complex

## Abstract

Accurate duplication and transmission of identical genetic information into offspring cells lies at the heart of a cell division cycle. During the last stage of cellular division, namely mitosis, the fully replicated DNA molecules are condensed into X-shaped chromosomes, followed by a chromosome separation process called sister chromatid disjunction. This process allows for the equal partition of genetic material into two newly born daughter cells. However, emerging evidence has shown that faithful chromosome segregation is challenged by the presence of persistent DNA intertwining structures generated during DNA replication and repair, which manifest as so-called ultra-fine DNA bridges (UFBs) during anaphase. Undoubtedly, failure to disentangle DNA linkages poses a severe threat to mitosis and genome integrity. This review will summarize the possible causes of DNA bridges, particularly sister DNA inter-linkage structures, in an attempt to explain how they may be processed and how they influence faithful chromosome segregation and the maintenance of genome stability.

## 1. Chromosome Mis-Segregation and Genome Instability

Chromosome mis-segregation is widely implicated in genomic instability diseases such as cancer [1]. In general, it is referred to as the appearance of incompletely separated chromosomes that manifest as bulky DNA bridges that remain connected, or alternatively, as the appearance of chromatin lagging between the daughter nuclear masses during anaphase. The former is usually termed as ‘anaphase bridges’ whereas the latter are referred to as ‘lagging chromosomes’ or ‘laggards’. In late 1930, Barbara McClintock proposed that anaphase bridges drive chromosome instability through a so-called breakage-fusion-bridge cycle mechanism [2,3]. She suggested that if inappropriate repair of DNA breaks occurs between chromatid arms, or dysfunctional telomeres of different chromosomes, this can lead to chromosomal fusion and the formation of dicentric chromosomes, which comprise two centromeres. In principle, if polar spindle attachment occurs independently at these two *in-cis* centromeres, there is 50% chance that a dicentric chromosome will produce two chromatid bridges during chromosome segregation. Alternatively, if chromosomal fusion occurs in between the broken sister arms or unprotected sister telomeres on the same chromosome, namely sister-chromatid fusion, it is almost certain that an anaphase bridge will be generated assuming that bipolar spindle attachment is not compromised. No matter which type of chromosomal fusion occurs, cells require a process of disconnection or cleavage of the resulting chromatin bridge during anaphase/telophase in order to complete the cell division cycle. It has been reported that persistently unresolved anaphase bridge formation can lead to cytokinesis failure and tetraploidization, particularly in the absence of Aurora B kinase, which was suggested to activate a NoCut checkpoint to allow additional time for anaphase bridge resolution [4,5]. How chromosomal bridges are resolved is still not fully understood but given the nature of the covalent linkage, their cleavage or rupture will inevitably create DNA breaks in the offspring cells, which is believed to promote further chromosomal re-fusion events (e.g., translocations) and genomic rearrangements. Studies in different organism models have revealed that bulky anaphase bridges may be resolved through different mechanisms. A study on budding yeast has suggested dicentric chromosomes are severed by the contraction of the actomyocin ring during cytokinesis [6]. However, in mammalian cells, it has been reported that anaphase bridges generated by telomere fusions can be cleaved by a cytoplasmic exonuclease, three prime repair exonuclease 1 (TREX1), after mitotic exit. The persistent chromatin bridge structures may prevent proper reformation of the nuclear envelop around the structures that allows TREX1 to digest and resolve the DNA bridge molecules. However, this in turn leads to a class of complex re-integration of DNA fragments, triggering intra- and inter-chromosomal rearrangements, known as chromothripsis [7]. On the other hand, a study using *Caenorhabditis elegans* embryos, interestingly, shows that anaphase bridges are resolved by a midbody-tethered endonuclease (LEM-3/Ankle1) at the late stage of mitosis [8,9]. Whether a similar midbody-dependent cleavage system also operates in other organisms requires further investigation as the midbody is not formed during early cell divisions in Drosophila embryos and yeasts have a closed mitosis that separates mitotic chromosomes from the cell cortex during cell division.

Apart from anaphase chromatin bridges, chromosome mis-segregation can also result from the failure of chromosome biorientation, which is mostly attributed to errors in microtubule organization and spindle-kinetochore assembly [10]. Sister kinetochores must connect to spindle microtubules emanating from the opposite poles (so-called amphitelic attachment) prior to chromosome disjunction, otherwise, sister chromatids will be distributed randomly into the daughter cells, driving aneuploidy [10]. Unattached or mis-attached chromosomes (e.g., monotelic and merotelic attachments) can lead to the formation of lagging chromosomes [11], which can subsequently turn into micronuclei if they fail to incorporate into the nuclei of the descendent cells. Furthermore, lagging chromatin can also be generated from broken chromosomes lacking centromeric regions and kinetochore structures. Interestingly, studies have shown that DNA inside micronuclei can undergo fragmentation, which promotes random nuclear integration and causes chromosome structural rearrangements in the succeeding offspring populations [12,13,14,15]. Therefore, chromosome mis-segregation not only can introduce numerical changes but also structural alterations of karyotypes—a common characteristic of cancer cells. Figure 1 depicts a brief summary of the cell division cycle, with the main focus being how accurate chromosome segregation occurs. Moreover, it provides a representation of how anaphase bridges and lagging chromosomes are generated, as well as, their pathological consequences.

For many decades, anaphase chromatin bridges and lagging chromosomes have served as a useful indicator to study chromosome mis-segregation activities. However, in the last decade, the discovery of a new form of DNA-bridging structure in anaphase cells termed ‘ultra-fine DNA bridges (UFBs)’ [16,17] significantly advances our understanding on the relationship between DNA replication and mitosis.

## 2. Classification of Ultra-Fine DNA Bridges

Ultra-fine DNA bridges are a form of stretched fine DNA linkage structures found in mammalian anaphase cells, which have escaped detection for many years because of their ‘DNA dye-proof’ property and poor nucleosome association. In 2007, two groups reported independently that a DNA translocase, *Plk1*-interacting checkpoint helicase (PICH), and a protein complex comprising Bloom’s syndrome helicase (BLM), TOP3A and RMI1 (so-called BTR complex, see below), respectively, exhibit a striking localization on thread-like structures linking separating chromatin masses in human anaphase cells [16,17]. These observations led to the proposal of the existence of an ‘invisible’ DNA linkage structure between the disjoined sister chromatids. Further labelling using bromodeoxyuridine (BrdU) incorporation confirmed that they are DNA structures [17]. Apart from PICH and BLM complex, it seems UFBs are also recognised by other nuclear factors as a very similar thread-like structure was previously described in a study of the cytological localization of a protein called RIF1. Although originally it was probably mis-interpreted as a cytoskeleton element in the midzone of anaphase cells [18], RIF1 has now been shown to be one of the UFB-associated components (see below) [19].

Different types of UFBs have now been described and classified based on where they arise and how they are induced [20,21,22]. It is generally believed that they include unresolved (1) double-stranded DNA catenanes, (2) late replication intermediates (LRIs) or (3) homologous recombination (HR) structures (Figure 2). The most common type of UFBs are those originating at centromeres (C-UFBs), also known as non-Fanconi Anemia (FA)-associated UFBs (non-FA UFBs) because of the lack of the FANCD2-FANCI protein (Fanconi Anemia complex) association (see below). Centromere-UFBs are prevalent in almost all early stage anaphase cells and are believed to be caused by the topological intertwinement of double-stranded DNA (dsDNA) catenanes that have yet to be resolved by topoisomerase IIα (TOP2A) [16,17,23]. Indeed, the amounts of non-FA UFBs are greatly increased in cells treated with low doses of TOP2A inhibitors, such as bisdioxopiperazines of the ICRF group [16,17,24,25], supporting the idea that unresolved catenation structures can induce UFBs. During unperturbed conditions, UFBs are predominantly found at centromeres and rarely on other chromosomal loci. A plausible explanation is that centromeric and pericentric DNA are preferentially embraced by cohesin, a giant molecular ring complex for sister chromatid cohesion [26], which could potentially prevent the completion of DNA decatenation. Indeed, suppression of the ‘prophase pathway’ that removes cohesin from chromosomal arms significantly induces UFB formation [27]. Another reason for the persistence of dsDNA catenanes, although unlikely, may be due to incomplete chromatin compaction at centromeric regions, as proper chromosome condensation is required for the removal of DNA catenanes [28,29]. It is worth emphasizing that although many experiments have shown that inhibition of DNA decatenation activity elevates UFB formation [16,17,24], this does not directly prove that centromeric UFBs are exclusively a product of catenated molecules. Given the highly repetitive nature of centromere sequences there remains a possibility that some of the C-UFBs contain HR structures.

Recently, studies have also suggested that dsDNA catenanes and the resulting UFBs can arise at ribosomal DNA regions (rDNA-UFBs) in chicken DT-40 cells [30,31]. This proposal is based on the findings that PICH knockout (PICH−/− DT-40 cells show elevated frequencies of anaphase DNA bridges at rDNA sites and PICH (even the ATPase-dead mutant) can stimulate TOP2A-mediated decatenation activity in vitro. Besides, more interestingly, PICH also forms a so-called ‘PICH-body’ with enriched TOP2A protein at rDNA loci on chicken mitotic chromosomes. The formation of the ‘PICH-body’ seems chicken cell-specific but it is intriguing why catenated structures preferably accumulate at rDNA loci when PICH is absent. A possible explanation could be that full rDNA condensation on chicken chromosomes may require extra decatenation action that is stimulated by the formation of a ‘PICH/TOP2A-body’. As PICH seems to have an interplay with TOP2A in the organization of chromosome arm architecture [32], the loss of PICH function (other than its proposed DNA-bridge resolution function) prior to anaphase may cause improper chromosome organization at rDNA regions, leading to the accumulation of DNA catenane intertwinements. However, it is again possible that some UFBs arising at rDNA loci may also result from HR intermediates, simply due to its highly repetitive nature, or from persistent RNA-DNA hybrids generated during rDNA transcription.

Apart from at centromeres and rDNA, UFBs can arise at ‘difficult-to-replicate’ genomic regions known as common fragile sites (CFSs), particularly under replication stress conditions, such as under mild treatment with the DNA polymerase inhibitor, aphidicolin [33]. However, unlike the C-UFBs and rDNA-UFBs, the ones found at CFSs are associated with the FANCD2-FANCI protein complex at their termini, thereby also known as FA-UFBs [25,34]. The formation of FA-UFBs is enhanced by the inhibition of the HR pathway [35,36]. Therefore, it is strongly believed that the FA-UFBs represent the incompletely replicated DNA intermediates that contain an under-replicated DNA region intertwining the flanking duplicated sister chromatids. The FANCD2-FANCI complex has been shown to participate in protecting stalled replication forks from degradation under replication stalling conditions [37]. The persistent association of FANCD2-FANCI complex at CFSs throughout mitosis may confer this DNA protective function. Alternatively, it is also possible that the chromatin-bound FANCD2-FANCI complex is trapped at CFSs during chromosome condensation and that prevents its dissociation during mitosis. Nevertheless, due to the incomplete replication nature, the resolution of the replication intermediate-associated UFBs is expected to generate DNA lesions, such as single-stranded DNA overhangs or gapped molecules after mitotic exit. In fact, the replication stress-induced DNA bridging lesions are transmitted into the successive G1 daughter cells and manifest as large nuclear bodies that are composed of 53BP1, a non-homologous end-joining factor [38,39]. Strikingly, the 53BP1-nuclear body structure persists throughout the entire G1 before their disassembly in the next S phase. It has therefore been suggested that the repair of the transgenerational DNA lesions may not start until the next S phase onset, while the nuclear bodies act as a damage shelter in G1 to prevent any illegitimate repair or rearrangements. Alternatively, the DNA breaks present in nuclear bodies may have been fixed in G1 but the removal of the huge ‘epigenetic mark’ requires S-phase specific factors such as components of the DNA replication machinery. Nevertheless, the function(s) of 53BP1-nuclear bodies in G1 cells remain(s) mysterious.

Another genomic locus where UFBs reside are telomeres (T-UFB) [40,41,42]. T-UFBs are not normally observed in unperturbed cells but seem to be specifically induced by over-expression of TRF2, a component of the Shelterin complex that protects telomeres. It is proposed that the formation of T-UFBs is not because of telomere-telomere chromosomal fusion as mentioned above, instead it is caused by replication stalling that generates incomplete replication intermediates mimicking the effects of replication stress [41]. However, aphidicolin treatment is not sufficient to trigger T-UFB structures. Since telomeres also contain highly repetitive sequences, it cannot be ruled out that the T-UFBs are also a product of unresolved homologous recombination intermediates that are generated during the rescue or repair of stalled forks. Thus, it would be interesting to examine if the Fanconi Anemia and 53BP1-nuclear bodies pathways are also activated at this specific genomic locus after TRF2 overexpression.

In addition to replication intermediate-induced UFBs, recently, two groups have reported the identification of HR-induced UFBs (HR-UFBs) in human cells, showing that these UFBs are not associated with FANCD2 protein and their formation relies on homologous recombination pathway activation. Importantly, both groups have established that HR-UFB structures seem very potent to generate chromosome damage during mitotic exit [35,36]. Chan et al. demonstrated that simultaneously inactivating two key Holliday junction resolvases, GEN1 and MUS81, leads to severe mitotic chromosome segmentation and the accumulation of anaphase UFBs [36]. On the other hand, Tiwari et al. found that partial inactivation of 53BP1, a negative regulator of the homologous recombination pathway, in human cancer cells, also elevates the formation of UFBs and more interestingly, the generation of signature chromosome rearrangements [35]. Furthermore, Tiwari et al. also mapped that the HR-UFBs induced in the 53BP1-depleted cells lead to chromosome rearrangements particularly at CFSs and centromeres, and further delineated a specific process named ‘sister chromatid rupture-bridging’ for their formation, which is distinct from breakage-fusion-bridge cycle-mediated genome rearrangements [35].

As discussed above, certain types of DNA entanglements are thought to preferentially arise at particular genomic loci, for instance, dsDNA catenanes at centromeres whereas replication intermediates arise at CFSs and telomeres. However, Tiwari et al. study highlights that a locus-based classification of UFBs may not truly reflect the nature of DNA entanglements, as HR-UFBs can also be found at both CFSs and centromeres [35]. Therefore, a multifaceted analysis is required to define the nature of UFBs, especially when they are induced after inactivation of factors involving DNA replication and repair. Alongside this, it is worth to note that the unresolved homologous recombination intertwinements can also lead to the formation of distinct types of chromatin bridges and laggards. Without both proper UFB and cytogenetic analyses, lagging chromosome/chromatin may be mis-interpreted as a result of impaired kinetochore-spindle attachments, rather than the accumulation of ultra-fine DNA entangling structures. Consequently, a comprehensive study of the features of UFBs provides us with a better understanding on the complicated process of chromosome segregation.

## 3. Ultra-Fine DNA Bridge-Associated Factors

The discovery of the UFBs was through immunofluorescent staining of its associated factors. Thus far the best well-known UFB associated factors are PICH translocase and BLM helicase. *Plk1*-interacting checkpoint helicase, also known as DNA excision repair protein ERCC-6-like (ERCC6L), belongs to the SNF2 family of ATPases and was described to interact with the mitotic kinase Polo-like kinase 1 (Plk1) through a ‘priming’ phosphorylation mechanism mediated by CDK1 [16]. Whereas PICH mostly localizes in the cytoplasm during interphase, it re-locates to centromeres/kinetochores and chromosome arms after nuclear envelope breakdown during mitosis and more strikingly it also decorates along the length of UFB structures throughout anaphase. PICH was initially proposed to act as an activator of the spindle assembly checkpoint. However, this proposed function was later proven to be due to an off-target effect of the PICH small interfering RNA (siRNA) oligos on the MAD2 checkpoint protein expression [43]. Nevertheless, consistent data demonstrates that PICH plays a critical role in chromosome architecture, organization and faithful chromosome segregation [16,32,44,45]. Knocking out PICH causes embryonic lethality in mice and it is proposed to be caused by excessive DNA damage generated during chromosome segregation [46]. Biochemical analyses show that PICH is a DNA translocase that is able to displace triplex DNA molecules and branch migrate a ‘4-way’ junction structure [47]. It has also been shown to remodel nucleosome positions [48]. However, this in vitro activity was not reproducible by another research group [47]. Recently, using single molecule analyses, PICH was found to exhibit a high binding affinity to duplex DNA especially when under stretching tension [47,49]. This probably provides a good explanation of why PICH can specifically associate to the DNA molecules of UFBs but not to the rest of DNA masses and also suggests that PICH may be the primary sensor of UFB structures. Indeed, PICH depletion abolishes the loading of many other known UFB-associated proteins including BLM and RIF1 (see below). Although PICH’s UFB binding property is getting clearer, its molecular action(s) on UFBs remains a mystery. A recent report has shown that PICH can stimulate TOP2A-mediated decatenation activity in vitro [30]. However, whether TOP2A is able to catalyze decatenation on stretched DNA structures requires further investigation.

Another prominent UFB-binding factor is BLM helicase. BLM is a 3′-5′ DNA helicase that belongs to the highly conserved family of RecQ helicases, which is known to be required for the maintenance of genome integrity. Along with BLM, most mammals possess four other RecQ genes that encode Werner syndrome protein (WRN), RECQ1, RECQ4, and RECQ5 [50,51]. However, apart from BLM, there is still no strong evidence to demonstrate that the other RecQ helicases also participate in processing UFBs in mitosis. In the absence of BLM, cells have defects in DNA replication [52,53] and elevated levels of homologous recombination (HR) activities, which lead to increased spontaneous chromosome breaks and sister chromatid exchanges [54,55]. Moreover, increased chromosome mis-segregation including the formation of anaphase bridges, lagging chromosome/chromatin, micronuclei formation and UFBs have also been reported [17,56]. BLM is well-known to physically and functionally interact with a variety of proteins for the maintenance of genome integrity. The most well-known partners are TOP3A [57,58], the RecQ-mediated genome instability protein complex 1 and 2 (RMI1 & 2) [59,60,61], replication protein A (RPA) [62,63], BRCA1 (Breast cancer susceptibility gene product) [64], and FA proteins [65,66].

TOP3A is a type IA topoisomerase, which has been shown to catalyze transient breaking and re-joining of single-stranded DNA (ssDNA) to relieve the torsional stress introduced into negatively supercoiled DNA when the complementary strands are separated by helicases [67,68,69,70]. When BLM acts in concert with TOP3A, they can mediate a reaction called double-Holliday junction dissolution, in which recombination structures are resolved through a co-action of branch migration of double-Holliday junctions, followed by decatenation of the resulting hemi-catenanes. This reaction generates exclusively non-crossover DNA products [71]. Actually, many of the cellular phenotypes of BS can be explained by the loss of the dissolution activity. Two additional proteins, RMI1 and RMI2, have been found to be essential for stabilizing the BLM complex [72,73], and they are also found on UFB structures [17]. Recently, it was identified that mutations in TOP3A and RMI1 also cause a Bloom’s syndrome-like disorder in humans [74]. The role of the BTR complex in dealing with entangled, branched DNA recombination intermediates may explain its mitotic role in chromosome segregation. The recruitment of the BTR complex to UFBs is PICH-dependent and is possibly mediated through the interaction between the C-terminus of PICH and BLM [48]. Interestingly, despite the strong interaction between BLM and TOP3A [58], a recent study suggested that they can be independently recruited to UFBs [49].

How the PICH and BLM complex function to resolve UFBs remains largely unclear in the field. However, the fact that TOP3A can act as ssDNA decatenase [69,70] and is able to resolve DNA substrates, such as late replication structures and dsDNA catenanes in the presence of BLM [49] suggests that their co-operative action may be critical for UFB resolution. Moreover, it is almost certain that UFBs are actively unwound by BLM helicase during anaphase, as the loading of RPA, an indicator of single-stranded DNA formation, is virtually abolished in the absence of BLM or its helicase activity [19,22,36]. It is conceivable that the generation of ssDNA regions on UFBs may allow efficient cleavage or even spontaneous rupture to occur as it is believed that ‘naked’ ssDNA tends to be more fragile than dsDNA. However, it remains undetermined if ssDNA molecules that are fully coated by RPA are as fragile as their ‘naked’ form. Alternatively, the loading of RPA may trigger a subsequent recruitment of other downstream effectors or nucleases that facilitates the resolution of the DNA bridging structures. Interestingly, the Fanconi Anemia protein FANCM translocase, was reported to localize to UFBs in a later stage of anaphase in a BLM-dependent manner [75]. However, whether this is dependent on RPA requires further investigation.

Moreover, two other proteins have been recently identified as potential UFB-associated factors—RIF1 (RAP1-interacting factor 1) and TOPBP1 (DNA Topoisomerase 2-binding protein 1). RIF1 was originally identified as a telomere associated protein [76] and was shown to have roles, acting downstream of ATM and 53BP1, in repairing DNA breaks [77,78] and controlling replication [79]. In mitosis, it has also been presented as an UFB-associated factor that may facilitate the UFB resolution [19]. Although RIF1 is identified as a component of the BLM complex and acts with BLM in the recovery of stalled replication forks [80], its recruitment to UFBs is independent of BLM helicase, although requires PICH. In vitro experiments show that RIF1 interacts with both the N- and C-terminus of PICH. However, deletions of these domains do not abolish RIF1’s loading onto UFBs in vivo. Thus, it is also postulated that RIF1’s recruitment may not be mediated through direct PICH interaction but instead may be triggered through the remodeling of UFB structures by PICH translocase activity [19]. Indeed, RIF1 itself has been found to bind directly to some branched DNA structures such as replication forks and 4-way junction substrates [80]. Alternatively, RIF1 may associate to UFBs via other PICH binding partners that are independent of BLM, such as TOP3A and RMI1/2.

Another protein that has been claimed to associate to UFBs is TOPBP1 (DNA topoisomerase 2-binding protein 1). TOPBP1 is known to be required to facilitate efficient initiation of DNA replication and cellular responses to DNA damage [81,82,83]. It has also been proposed to function during mitotic progression because of its striking localization on centrosomes [84] and focal formation on mitotic chromosome arms in chicken DT40 and human cancer cells [85]. Besides, it also exhibits a UFB-like localization in anaphase [86]. However, different from other known UFB-associated factors, TOPBP1 seems to appear only on a sub-set of and in a short sub-region of UFB structures. More surprisingly, its localization is independent of both BLM and PICH proteins [87], where the latter as mentioned above is believed to be the primary UFB-binding protein recruiter. Given that TOPBP1 forms foci on chromatin prior to the appearance of UFBs during anaphase, it is possible that TOPBP1, like the FANCD2-FANCI complex, may act to ‘pre-mark’ some problematic or damaged regions that subsequently give rise to a different form of DNA bridging structures. Therefore, unlike other bona fide UFB-associated factors, TOPBP1 may start to function earlier during mitosis. Indeed, it has been shown to recruit SLX4 [85], a key component for structure-specific endonuclease assembly, which may facilitate the cleavage of branched or entangled DNA molecules on chromosomes. Structural studies reveal that TOPBP1 contains eight BRCT domains, suggesting a ‘scaffold-like’ role for protein-protein recruitment [88]. Recently, it has been found that the seventh and eighth BRCT domains of TOPBP1 are required to interact with TOP2A and thus suggests a role in the recruitment of TOP2A to UFBs for DNA decatenation during mitosis [87]. Although an interaction was observed in vitro, TOP2A is always poorly detected along the length of UFBs and can only be visualized as punctuate foci on the UFB-like structures by proximity ligation assay [87]. Thus, it seems that the recruitment of TOP2A by TOPBP1, if it occurs, is unlikely merely mediated through their direct interaction. It may be also influenced by localized DNA structures. A summary of the assembly of these UFB-binding factors is presented in Figure 3.

Malfunctions of UFB-binding factors have been described to elevate the formation of UFBs, which supports the idea of their potential role in the UFB resolution pathway during mitosis. However, it should be emphasized that all of the UFB-binding factors, except PICH translocase, also have important functions during DNA replication, recombination and repair in interphase cells. Therefore, caution should always be taken when interpreting phenotypes of chromosome mis-segregation in these mutants, as they may not necessarily reflect solely the loss of their potential UFB-resolution activities. Instead it could also be as a consequence of increased formation and accumulation of abnormal DNA intertwinements arising during S phase.

## 4. How Are Different Types of Ultra-Fine DNA Bridges Resolved?

It is becoming clearer what kind of DNA intertwining structures can give rise to UFBs. Nevertheless, it remains a big question in the field yet to be answered about how these entanglements are properly resolved during mitosis and what their impacts on genome integrity are. As discussed above, UFBs in general can be categorized as persistent (1) dsDNA catenanes, (2) late replication intermediates or (3) homologous recombination structures. Given their structural differences, it is reasonable to think that distinct resolution pathways may be employed for their individual resolution. However, regardless of their origins, it seems that all UFBs are recognized by both PICH and BLM complex, which may indicate that this complex probably plays a key role to activate (different) resolution pathways. To resolve dsDNA catenation-induced UFBs, TOP2A is believed to be the most suitable candidate. It is well-known that TOP2A is a potent enzyme to catalyze decatenation of dsDNAs [89]. Its enrichment at centromeres during mitosis suggests that residual topological DNA linkages may persist. Under normal circumstances, TOP2A is probably fully capable of resolving most, if not all, catenaned DNA molecules. However, it is unclear if the same reactivity occurs on the catenated structure that are under strong tension, such as in the situation of anaphase UFB formation. The stretching condition may hinder or reduce TOP2A’s DNA binding or decatenation activity. Therefore, it is conceivable that PICH/BLM complex serves as a co-activator, or even as an alternative decatenase in order to resolve ‘stretched DNA catenation molecules’ refractory to the cleavage of TOP2A. Further single molecule experiments will help to address this possibility. Nevertheless, it is almost certain that the decatenation or resolution process is extremely fast in vivo, as virtually all UFBs arising between sister centromeres disappear within minutes after anaphase onset [16].

In contrast to full dsDNA catenanes that topologically inter-link sister DNAs, late replication intermediates (LRIs) intertwine sister chromatids through their under-replicated DNA region. Because DNA replication has not been completed, the resolution or cleavage of LRI structures will inevitably create DNA breaks or ssDNA regions, which can pose a severe threat to genome integrity. Figure 4 depicts two possible scenarios that cells may utilize in order to disjoin LRI-induced UFBs during mitosis. The first one does not involve cleavage of the branched DNA molecules, instead it requires TOP3A to disjoin the ssDNA hemi-catenanes that are generated by BLM-mediated unwinding or spindle pulling action. It is possible that TOP1 may also be involved to relieve the torsional stress [90] created during the separation of the DNA strands during anaphase. The second pathway requires ‘DNA pre-cleavage’ by structure-specific nucleases. In fact, studies have shown that SMX tri-nucleases comprising of SLX1-4, Mus81-EME1 and XPF-ERCC1 [91] are located at the chromosomal sites; e.g., CFSs, where are the hotspot of the formation of the FA-associated UFBs [92]. It may be argued that DNA cleavage at stalled forks could potentially lead to fork collapse and DNA fragmentation [93]. However, if it occurs after chromosome condensation, the broken chromatid arms could remain held and tethered together [94,95]. In support of the cleavage model, Mus81 has been shown to promote common fragile site expression in human mitotic cells [96]. On the other hand, the presence of the FANCD2-FANCI complex, as discussed above, may protect the DNA breaks from excessive processing or resection until anaphase onset. During chromosome disjunction, the last step to resolve the resulting UFB is to unwind the partial DNA duplex into ssDNA. This model could explain why the PICH and BLM complex always decorate along the length of UFBs whereas FANCD2-FANCI complex locate at their termini. No matter which model is involved, active BLM helicase activity seems to be critical.

Lastly, how might HR-mediated UFBs be resolved? HR structures are considered as a type of covalently linked DNA entanglement between sister chromatids. It has been suggested that there are two pathways for their resolution. Similarly, one requires BTR complex but without the involvement of DNA nucleases, which is called double-Holliday junction dissolution. The second pathway is dependent on SMX tri-nucleases or a HJ-specific endonuclease, GEN1/YEN1 and requires DNA cleavage [91,97]. It is believed that most of the HJ structures are resolved during interphase via the double-Holliday junction dissolution pathway. The residual HR structures are then cleaved by SMX tri-nucleases and GEN1 during mitosis [98]. These two different enzymatic reactions most likely provide a sufficient clearance capacity to ensure sister chromatids are free of homologous recombination inter-linkage structures that otherwise hinder normal chromosome disjunction. The fact that cells evolve multiple ways to resolve homologous recombination intermediates may indicate that these structures are highly problematic or toxic. Indeed, two recent studies have found that the accumulation of homologous recombination structures can lead to severe chromosome mis-segregation during mitosis and genome rearrangements in offspring cells. Although HR-UFBs are also recognized by PICH and the BTR complex, it seems that they are inefficient to remove them properly. In parallel, the HR-UFBs induced in the 53BP1-depleted cells also result in characteristic rupture of sister chromatids. Therefore, these studies indicate that PICH and the BTR complex may not be designed to deal with HR-mediated DNA bridging molecules in mitosis.

## 5. Conclusions

The discovery of UFBs provides an important link between the activities of DNA replication and repair throughout interphase, to chromosome segregation during mitosis. It also highlights that many DNA metabolism factors in fact contribute extremely important functions outside of interphase in order to facilitate mitosis. Moreover, the specific signatures of DNA lesions associated with different types of UFBs also provide us with a new direction to the understanding of how complex gross chromosome rearrangements may arise and evolve in tumor cells. Although it remains a challenge to understand how cells deal with UFB-linkage structures during mitosis and how the resulting DNA lesions impact genome integrity in the successive cell generations, future work will definitely provide further valuable knowledge on the mechanisms of genome stability maintenance. In addition, future studies should not be limited to the UFB resolution pathways but also include how the formation of these potentially dangerous DNA intertwining molecules is suppressed and avoided.

## Figures and Tables

**Figure 1 genes-09-00623-f001:**
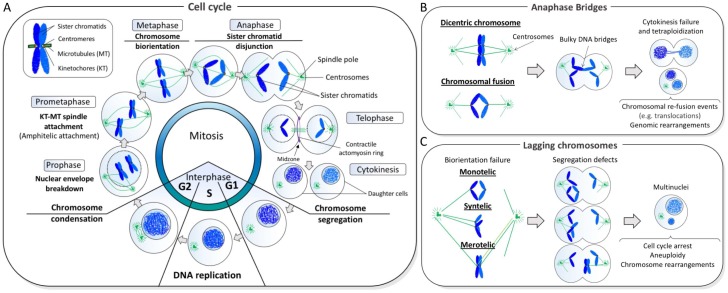
Normal and abnormal chromosome segregation. (**A**) In each cell division cycle, sister chromatid cohesion is established in S phase while DNA is being replicated. As cells enter mitosis, the duplicated DNAs begin to condense through a process driven by the condensin complex. During prometaphase-metaphase progression, kinetochores resided at the centromeric regions of chromatids attach to microtubules that emanate from the opposite centrosomes, leading to the alignment of chromosomes at the cell equator. Once every single chromosome is biorientated, the disjunction of sister chromatids occurs at anaphase that ensures equal distribution of the chromosomes into two daughter cells. (**B**) An abnormal fusion between sister chromatids generates bulky anaphase bridge formation. If chromatin bridges are not resolved in anaphase/telophase, they can impair cell division and leads to binucleated cells formation. On the other hand, if improper resolution/cleavage occurs, this can cause chromosomal breakage and that predisposes to re-fusion events, leading to genomic rearrangements; (**C**) Failure to establish chromosome biorientation (achieved by amphitelic kinetochore-spindle attachment) can lead to imbalance chromosome transmission, leading to aneuploidy and promoting chromosome rearrangements. Spindle attachment errors include microtubule connection on single kinetochore (monotelic), attachment of two sister kinetochores from the same spindle pole (syntelic) and attachment of a single kinetochore to microtubules emanating from two spindle poles (merotelic).

**Figure 2 genes-09-00623-f002:**
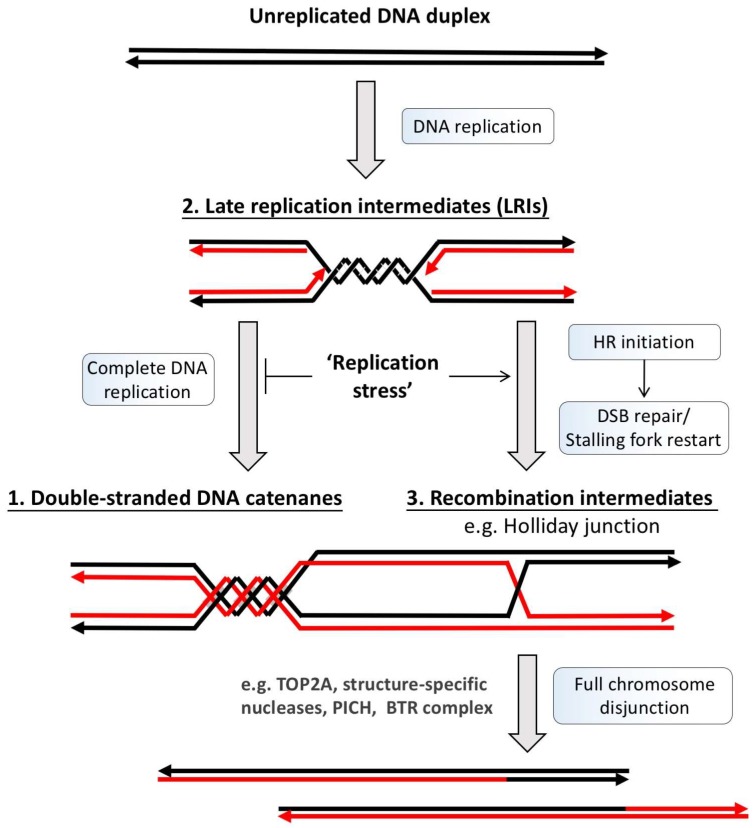
DNA metabolism generates potential DNA linkage structures intertwining sister DNA strands. DNA duplication is achieved through a semiconservative replication process in which the unreplicated DNA duplex template (black lines) is first separated, followed by the generation of two new DNA strands by DNA polymerases (red lines). Due to the helical intertwining nature of the DNA duplex, its unwinding produces supercoiling, which is subsequently converted into double-stranded DNA catenation after replication (1). However, if the completion or termination of DNA replication fails, such as under replication stress conditions, the under-replicated DNA regions can also intertwine sister chromatids. These regions usually arise at late replication sites and manifests as double stalled forks or replication intermediates (2). On the other hand, the rescue of the stalled or damaged replication forks by homologous recombination (HR) reactions can also lead to sister chromatid intertwinements via the generation of HR intermediates, such as D-loop and Holliday junctions (3). Prior to or during mitosis, cells have to resolve all of these topological and covalent DNA linkages to allow faithful sister chromatid disjunction to proceed. BTR: Bloom helicase/Topoisomerase IIIα/RecQ-mediated genome instability protein complex 1 and 2; DSB: Double-stranded break; TOP2A: topoisomerase IIα; PICH: *Plk1*-interacting checkpoint helicase.

**Figure 3 genes-09-00623-f003:**
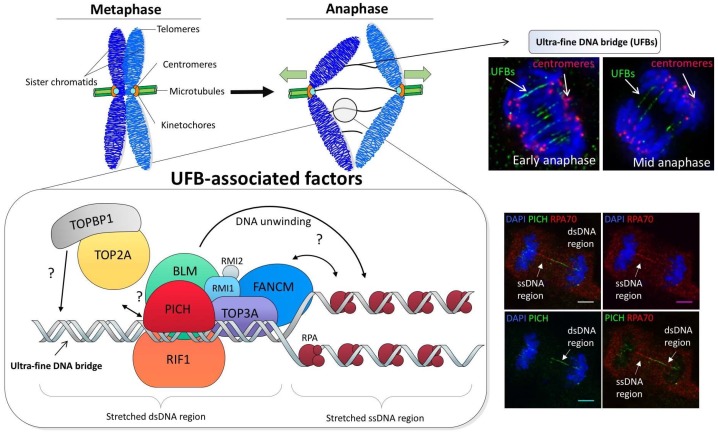
A proposed model of the assembly of the ultra-fine DNA bridge (UFB)-associated factors on UFB structures. (**Top**) Schematic representation of the formation of ultra-fine DNA bridges, which connect sister chromatids together in anaphase. The right image panel shows early and mid anaphase cells displaying ultra-fine DNA bridges (unpublished images). (**Bottom**) At the onset of anaphase, DNA molecules that intertwine sister chromatids are under the spindle pulling tension. The stretching forces may cause DNA conformational changes and hence initiates the binding of PICH translocase to the double-stranded DNA (dsDNA) region of the UFB. The BLM/TOP3A/RMI1-2 (BTR) complex and RIF1 protein are also recruited, respectively, probably by protein-protein interactions with PICH through its different domains. The BLM helicase subsequently unwinds the DNA duplex into single-stranded DNA (ssDNA), which allows the loading of replication protein A (RPA). In parallel, FANCM translocase is also recruited in a BLM-dependent manner. A sub-region of a sub-set of UFB structures is recognized by TOPBP1 in a PICH and BLM-independent manner, which may recruit TOP2A to a specific site of UFBs for decatenation. The right image panel shows the differential localization of PICH translocase and RPA70 to the dsDNA and ssDNA regions of a UFB, respectively, in an anaphase cell of GM00637 normal diploid fibroblast, scale bar; 5 μm (unpublished images).

**Figure 4 genes-09-00623-f004:**
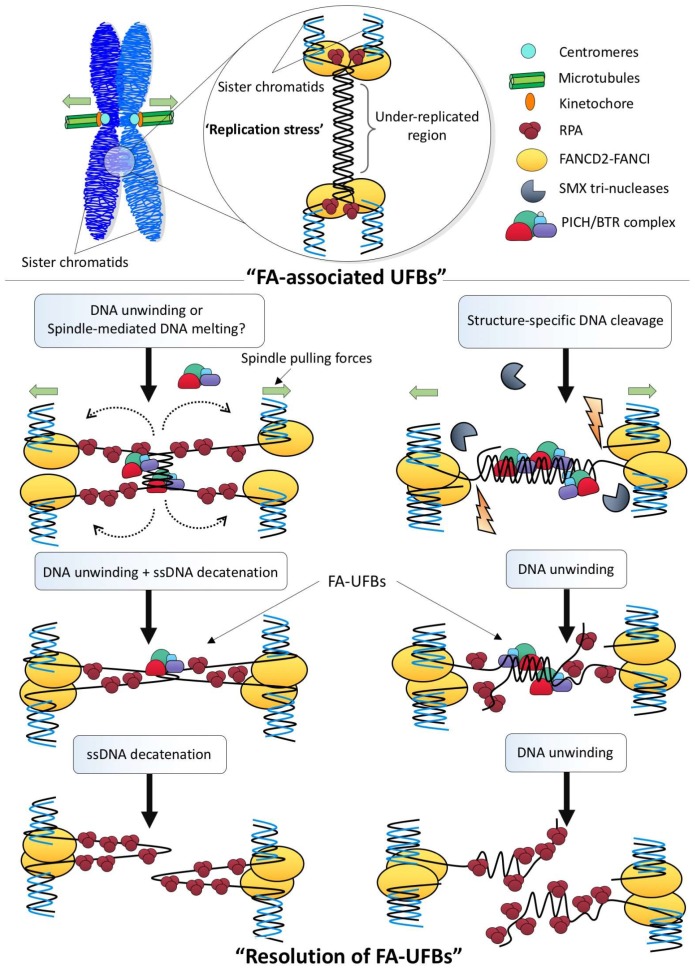
Possible resolution pathways to disjoin late replication intermediate structures during mitosis. Under replication stress conditions, cells may fail to complete DNA replication prior to mitosis, which leaves under-replicated regions. The FANCD2-FANCI complex is then recruited to these sites, most likely to rescue or stabilize the stalled forks. During sister chromatid disjunction, the under-replicated region may be converted into the ultra-fine DNA bridge structure, possibly either through the melting of the under-replicated DNA duplex by the spindle pulling forces or via the unwinding by BLM helicase activity (**Left**). As a result, a structure containing both single-stranded DNA and positive supercoiling is produced, which can turn into hemi-catenane structures that can be subsequently resolved by TOP3A. Alternatively, as under-replicated structures contain branched junctions between the fully and under-duplicated regions, structure-specific nucleases (e.g., SMX tri-nucleases) may be employed to first cleave the DNA linkages prior to or during chromosome segregation (**Right**). The separation of the cleaved sister chromatids will then produce a UFB structure comprising mainly DNA duplex of the under-replication region. The resolution of the DNA bridge linkage can be mediated by the BLM-dependent unwinding reaction without the involvement of ssDNA decatenation. However, the former pathway will produce single-stranded DNA gaps whereas the latter will generate single-stranded overhangs structures. BTR: Bloom helicase/Topoisomerase IIIα/RecQ-mediated genome instability protein complex 1 and 2; FA: Fanconi Anemia; FANCD2-FANCI: Fanconi Anemia complex; RPA: Replication protein A; PICH: *Plk1*-interacting checkpoint helicase; BTR: SMX: SLX1-4, Mus81-EME1 and XPF-ERCC1.

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
