# Peer review of "The Unresolved Problem of DNA Bridging"

_genes, 2018, doi:10.3390/genes9120623_

Round 1

Reviewer 1 Report

Review of Fernandez-Casanas and Chan

This is a comprehensive and detailed review of chromosome bridges in mitosis, the different types and how they are formed, the biochemistry involved, and their resolution and effects on genome instability. Overall, this is a thorough review that covers a lot of material, and includes three useful and informative figures. In some instances it becomes very dense in details and somewhat hard to follow.

In particular, some parts of the manuscript go from one idea to another rather rapidly, and in these cases it becomes more difficult to draw connections and understand the broad picture. Some examples are the top of page 2, which appears to jump between ideas (and model organisms); better connecting the different points would be useful. Also in this part, “leads to chromothripsis”- more details might be useful to fully understand this. Also, if TREX1 is an exonuclease how can it cleave within a DNA molecule? Another example is the bottom of some page, first paragraph of part 2- also jumps between ideas (e.g. RIF1) rapidly with minimal explanation.

While the manuscript is overall well written, there are some cases of minor grammatical and typographical errors, so I would recommend a careful english edit by the authors before submission of a final version. 

The most important such case is the title of the paper: “The Unresolved DNA bridging Problem” what is unresolved, the bridging or the problem? Suggest to change to either “the unresolved problem of DNA bridging” or “the problem of unresolved DNA bridging”.

Another important case is section 4 title “ How different ultra-fine DNA bridges are resolved?” should be “how are…” or, alternatively, remove the question mark.

Some other random examples: line 146 “ inhibition of the HR pathway does not diminish, instead enhance”- should be enhances

line 48 “ In mammalian cells, it has reported”

line 338 “ TOP2A is probably believed” 

These are minor comments. Overall I find this to be a useful review. 

Author Response

This is a comprehensive and detailed review of chromosome bridges in mitosis, the different types and how they are formed, the biochemistry involved, and their resolution and effects on genome instability. Overall, this is a thorough review that covers a lot of material, and includes three useful and informative figures. In some instances it becomes very dense in details and somewhat hard to follow.

In particular, some parts of the manuscript go from one idea to another rather rapidly, and in these cases it becomes more difficult to draw connections and understand the broad picture. Some examples are the top of page 2, which appears to jump between ideas (and model organisms); better connecting the different points would be useful. Also in this part, “leads to chromothripsis”- more details might be useful to fully understand this. Also, if TREX1 is an exonuclease how can it cleave withina DNA molecule? Amended and included a short description.  Another example is the bottom of some page, first paragraph of part 2- also jumps between ideas (e.g. RIF1) rapidly with minimal explanation. Amended

While the manuscript is overall well written, there are some cases of minor grammatical and typographical errors, so I would recommend a careful english edit by the authors before submission of a final version. 

The most important such case is the title of the paper: “The Unresolved DNA bridging Problem” what is unresolved, the bridging or the problem? Suggest to change to either “the unresolved problem of DNA bridging” or “the problem of unresolved DNA bridging”. Amended.

Another important case is section 4 title “ How different ultra-fine DNA bridges are resolved?” should be “how are…” or, alternatively, remove the question mark. Corrected.

Some other random examples: line 146 “ inhibition of the HR pathway does not diminish, instead enhance”- should be enhancesCorrected.

line 48 “ In mammalian cells, it has reported” Corrected.

line 338 “ TOP2A is probably believed” Corrected. 

Reviewer 2 Report

The manuscript by Fernández-Casañas and Chan summarizes the current knowledge on the biology of ultrafine DNA bridges (UFBs). Overall, it is a well-organized and timely review covering a classification of the different types of UFBs, how they are formed and resolved, and the factors that associate with them. The review will be of interest to those studying genome integrity maintenance mechanisms. 
The reviewer understands that it is difficult to describe the complex DNA transactions involved during UFB formation and resolution, but the writing is a bit heavy and complex. The manuscript would benefit from a careful round of polishing. 

Comments and suggestions aimed at improving and polishing the manuscript are listed below for consideration. 
1) Lines 29 and 30: CIN and BFB acronyms are not needed. In general, there are too many acronyms in the manuscript which complicates the reading at times. Only keeping UFB related acronyms would help. 
2) Line 48: it “has” been reported 
3) Line 76: remove “Ultra-fine DNA bridges” and only keep the acronym as it has been defined on line 73. 
4) Line 86: has been shown “to be” a component 
5) Line 103: remove “overall” 
6) P 3 Figure 1: about the two intermediates in the middle of the figure i.e. the single-stranded DNA hemi-catenates and its decatenation. It is difficult to understand how they are formed and resolved. Is there another way to represent this? 
7) Line 118: “under-replicated” 
8) Line 119: “stalled” 
9) Line 120: “stalled” 
10) Lines 117-120: sentence is too long and too complicated 
11) Line 142: CFS and RS acronyms not necessary 
12) Line 143: mild treatment “with” 
13) Lines 145-148: Long complicated sentence. Rephrase 
14) Lines 153-156: Long and complicated sentence. Rephrase 
15) Line 157: NB acronym not necessary 
16) Line 162: “such as components of the DNA replication machinery” 
17) Line 166: expression “of TRF2, a component of the Sheltering complex that protects telomeres” 
18) Line 169: remove “in general” 
19) Lines 174-198: This section on HR-UFBs needs to be improved. It is difficult to understand because there are too many concepts involved 
20) Line 199: Better title: “Factors that associate with ultrafine DNA bridges”? 
21) Line 199: ultra-fine or ultrafine. Not consistent in the manuscript 
21) Line 200: The discovery of UFBs 
22) Line 206: NEBD acronym not needed 
23) Line 208: SAC acronym not needed 
24) Line 209: “was later proven to be due” 
25) Line 214: “PICH is a DNA translocase that is able to displace triplex DNAs and branch migrate” 
26) Line 215: It “has” also been shown 
27) Line 219: “associate to DNA in UFBs but not to” 
28) Line 226: (Bloom’s syndrome, BS) 
29) Line 232: “activities, which manifested as…” 
30) Line 233: SCEs acronym not necessary 
31) Line 253: is possible “mediated” through 
32) Lines 254-255: incorrect sentence. 
33) Line 257: and is able to 
34) Line 287: mitotic 
35) Line 320: reference for the images if already published? 
36) Line 331: it remains a big question in the field yet to be answered how
37) Line 342: catenanes molecules 
38) Line 346: “stretched catenated DNA molecules” 
39) Line 355: apply use 
40) Line 363: stalling stalled 
41) Line 372: is involved 
42) Figure 3: it is difficult to understand the direction of the forces/tensions 
43) Line 377: and that leaves 
44) Line 378: stalling stalled 
45) Line 378: chromatids 
46) Line 380: by the spindle pulling forces 
47) Line 385: nucleases 

Author Response

Casañas and Chan summarizes the current knowledge on the biology of ultrafine DNA bridges (UFBs). Overall, it is a well organized and timely review covering a classification of the different types of UFBs, how they are formed and resolved, and the factors that associate with them. The review will be of interest to those studying genome integrity maintenance mechanisms. 
The reviewer understands that it is difficult to describe the complex DNA transactions involved during UFB formation and resolution, but the writing is a bit heavy and complex. The manuscript would benefit from a careful round of polishing. We have now included a new figure and amended the old figures to help the text description.

Comments and suggestions aimed at improving and polishing the manuscript are listed below for consideration. 
1) Lines 29 and 30: CIN and BFB acronyms are not needed. In general, there are too many acronyms in the manuscript which complicates the reading at times. Only keeping UFB related acronyms would    help. Corrected.We are now using full form of the names and just keeping the UFB related acronyms only. 
2) Line 48: it “has” been reported. Corrected.
3) Line 76: remove “Ultra-fine DNA bridges” and only keep the acronym as it has been defined on line 73. Corrected.
4) Line 86: has been shown “to be” a component.Corrected.
5) Line 103: remove “overall”. Corrected.
6) P 3 Figure 1: about the two intermediates in the middle of the figure i.e. the single-stranded DNA hemi-catenates and its decatenation. It is difficult to understand how they are formed and resolved. Is there another way to represent this?Corrected.

7) Line 118: “under-replicated”. Corrected.
8) Line 119: “stalled”. Corrected.
9) Line 120: “stalled”. Corrected.
10) Lines 117-120: sentence is too long and too complicated.Corrected.
11) Line 142: CFS and RS acronyms not necessary.Corrected.
12) Line 143: mild treatment “with”. Corrected.
13) Lines 145-148: Long complicated sentence. Rephrase.Corrected.
14) Lines 153-156: Long and complicated sentence. Rephrase.Corrected.
15) Line 157: NB acronym not necessary. Corrected.
16) Line 162: “such as components of the DNA replication machinery”.Corrected.
17) Line 166: expression “of TRF2, a component of the Sheltering complex that protects telomeres”.Corrected.
18) Line 169: remove “in general”. Corrected.
19) Lines 174-198: This section on HR UFBs needs to be improved. It is difficult to understand because there are too many concepts involved. Corrected.
20) Line 199: Better title: “Factors that associate with ultrafine DNA bridges”?.Corrected to Ultra-fine DNA bridge (UFB)-associated factors.
21) Line 199: ultra-fine or ultrafine. Not consistent in the manuscript.All are changed to ultra-fine. 
21) Line 200: The discovery of UFBs. Corrected.
22) Line 206: NEBD acronym not needed. Corrected.
23) Line 208: SAC acronym not needed. Corrected.
24) Line 209: “was later proven to be due”.Corrected.
25) Line 214: “PICH is a DNA translocase that is able to displace triplex DNAs and branch migrate”.Corrected.
26) Line 215: It “has” also been shown. Corrected.
27) Line 219: “associate to DNA in UFBs but not to”.Corrected.
28) Line 226: (Bloom’s syndrome, BS). Corrected.
29) Line 232: “activities, which manifested as…”.Corrected.
30) Line 233: SCEs acronym not necessary. Corrected.
31) Line 253: is possible “mediated” through. Corrected.
32) Lines 254-255: incorrect sentence. Corrected.
33) Line 257: and is able to. Corrected.
34) Line 287: mitotic. Corrected.
35) Line 320: reference for the images if already published? Corrected:(Unpublished images).
36) Line 331: it remains a big question in the field yet to be answered how.Corrected.
37) Line 342: catenanes molecules. Corrected.
38) Line 346: “stretched catenated DNA molecules”.Corrected.
39) Line 355: apply use. Corrected.
40) Line 363: stalling stalled. Corrected.
41) Line 372: is involved. Corrected.
42) Figure 3: it is difficult to understand the direction of the forces/tensions.Corrected.
43) Line 377: and that leaves. Corrected.
44) Line 378: stalling stalled. Corrected. 
45) Line 378: chromatids. Corrected. 
46) Line 380: by the spindle pulling forces.Corrected.
47) Line 385: nucleases. Corrected.